# A Delaunay Edges and Simulated Annealing-Based Integrated Approach for Mesh Router Placement Optimization in Wireless Mesh Networks

**DOI:** 10.3390/s23031050

**Published:** 2023-01-17

**Authors:** Tetsuya Oda

**Affiliations:** Department of Information Engineering, Okayama University of Science (OUS), Okayama 700-0005, Japan; oda@ous.ac.jp

**Keywords:** Wireless Mesh Networks, intelligent algorithms, simulated annealing, Delaunay Edges, mesh router placement, optimization, disaster environment

## Abstract

Wireless Mesh Networks (WMNs) can build a communications infrastructure using only routers (called mesh routers), making it possible to form networks over a wide area at low cost. The mesh routers cover clients (called mesh clients), allowing mesh clients to communicate with different nodes. Since the communication performance of WMNs is affected by the position of mesh routers, the communication performance can be improved by optimizing the mesh router placement. In this paper, we present a Coverage Construction Method (CCM) that optimizes mesh router placement. In addition, we propose an integrated optimization approach that combine Simulated Annealing (SA) and Delaunay Edges (DE) in CCM to improve the performance of mesh router placement optimization. The proposed approach can build and provide a communication infrastructure by WMNs in disaster environments. We consider a real scenario for the placement of mesh clients in an evacuation area of Kurashiki City, Japan. From the simulation results, we found that the proposed approach can optimize the placement of mesh routers in order to cover all mesh clients in the evacuation area. Additionally, the DECCM-based SA approach covers more mesh clients than the CCM-based SA approach on average and can improve network connectivity of WMNs.

## 1. Introduction

Wireless Mesh Networks (WMNs) can build a communications infrastructure using only routers (called mesh routers), making it possible to form networks over a wide area at low cost [1,2,3]. They also can act quickly and extensively to restore the networks even when the communication environment is cut off by a disaster in a wide area. WMNs can be used not only in such environments, but also for Internet of Things (IoT) applications [4]. For instance, they can be used for communication between robots in an indoor factory environment. Therefore, WMNs are expected to be utilized as a communication infrastructure for edge systems in the IoT era. However, the placement of mesh routers significantly affects the network operations, including installation cost, communication range, and maintenance.

The placement of mesh routers is a multi-objective optimization problem [5] and is an NP-hard [6] problem because it requires the optimization of both the connectivity between mesh routers and the coverage of mesh clients by mesh routers. Therefore, much research is being carried out to optimize the placement of mesh routers in order to improve the performance of wireless communication through WMNs. There are many meta-heuristics [7,8], such as Genetic Algorithms (GA) [9], Hill Climbing (HC) [10], Simulated Annealing (SA) [11], Tabu Search (TS) [12], and Particle Swarm Optimization (PSO) [13], that can be used for the optimization of mesh router placement.

In this paper, we present a Coverage Construction Method (CCM) for the optimization of mesh router placement. The CCM performs the computation until all mesh routers are connected considering the maximum communication range of mesh routers. In addition, we propose a mesh router placement optimization approach that combines SA and Delaunay Edges (DE) [14] in CCM (called DECCM-based SA). The DECCM-based SA approach considers a more realistic placement of mesh clients. In DECCM-based SA, the DE is used to cover the spread of mesh clients in considered areas. We compare the proposed DECCM-based SA approach with the CCM-based SA approach without Delaunay edge. In the simulation, we consider two metrics for the evaluation of the proposed approaches the Size of Giant Component (SGC) [15] and the Number of Covered Mesh Clients (NCMC). The SGC metric is an indicator for evaluating the connectivity between mesh routers, and the NCMC metric is for the coverage of mesh clients by mesh routers. We present a visualization of the results after the optimization of mesh routers. In addition, we consider the Geographic Information System (GIS) and hazard maps to develop an optimization scenario for mesh router placement in a disaster environment.

The contributions of this research work are as follows:Implementation of CCM for mesh router placement optimization.Implementation of a mesh router placement optimization system by combining DE and SA.Comparison of the proposed mesh router placement approaches and evaluation of their impact on connectivity and coverage.Performance evaluation of the proposed mesh router placement optimization system in a real environment using GIS.Construction of simulation scenarios considering evacuation centers based on GIS data assuming a disaster environment.

The paper is organized as follows. The related work is presented in Section 2. In Section 3, we describe the mesh router placement problem. In Section 4, we discuss the proposed approach. In Section 5, we present the simulation results. Finally, conclusions and future work are given in Section 6.

## 2. Related Work

In this section, we present the related work for the mesh router placement problem. There are many research works for optimizing mesh router placement. The communication performance of WMNs is affected by connectivity, coverage, cost, load-balancing, throughput, delay, capacity, and interference [16]. In [17], the authors considered Quality of Service (QoS) requirements by optimizing the placement of mesh routers, prioritizing topologies with less interference. In [18], the authors considered a load balancing arrangement of gateways in the WMNs.

Since the mesh router placement problem is NP-hard, meta-heuristics are often applied to mesh router optimization approaches. In [19], the authors presented the Non-Dominated Sorting Genetic Algorithm (NSGA-II) [20]. They stated that the main cost in the WMN is the placement and management of the Internet Access Points (ITAPs) used to connect to the Internet. Therefore, the objective is to minimize the number of required ITAPs while maximizing the traffic that can be served to each unit. In [21,22], the authors considered SA to solve the static/dynamic mesh router node placement problem. In [21], they assumed a static environment and used service priorities as an evaluation metric for optimizing mesh router placement. In [22], they assumed a mobile environment and that the performance indicators for mesh router placement optimization are connectivity and coverage, as well as average travel distance by routers. In [23], the authors compared the performance of SA and GA for mesh router placement optimization while in [24] they compared the performance of different types of GAs (NSGA-II and the Multi-Objective Genetic Algorithm (MOGA) [25]). In [23], the objective of mesh router placement optimization was to minimize the network cost of the WMN while satisfying the QoS. In [24], the authors considered as evaluation metrics for optimization coverage, reliability, and cost. They showed that the NSGA-II has better performance than MOGA for hypervolumes.

There are also some approaches for mesh router placement optimization that combine several meta-heuristics, clustering algorithms, or other methods. In [26], the authors proposed Multi-objective Simulated Annealing based Centre of Mass (MCM) as a mesh router placement optimization method and Multi Objective Simulated Annealing based Centre of Mass (MSAC), which combines SA and MCM. They showed that MSAC achieves a better tradeoff between WMN coverage and cost because it does not consider min–max regret values for many large instances and in most cases provides a good Pareto front. In [27], a combined GA and clustering method (GA-CL) was proposed as a mesh router placement method that considers the dynamic location of all mesh routers and indoor obstacles. This method can minimize interference among mesh routers. The GA-CL was shown to improve the performance by about 40% compared to the random selection method. In the Non Line-of-Sight (NLoS) environment, the GA-CL improves the performance by about 50% compared with the Line-of-Sight (LoS) environment.

In many approaches for mesh router placement optimization, the real-world mesh router placement target is abstracted by a grid or other means. In some cases the mesh router placement is performed assuming the real environment. In [28], the authors considered mesh router placement optimization for a real environment. They proposed a mesh router placement optimization method for Wanglang National Nature Reserve in Sichuan province, China. In [29], the authors considered the placement optimization of Unmanned Aerial Vehicles (UAVs) equipped with ad hoc communication capabilities for providing WMN services. The UAV placement optimization system was evaluated using a model based on GIS data from the Swedish Government Agency. In the GIS data, three cities with different areas and building structure densities were selected.

## 3. Mesh Router Placement Problem

In this section, we describe the mesh router placement problem. For most formulations, node placement problems are shown to be computationally hard to solve to optimality [30,31,32,33,34], and therefore heuristic and meta-heuristic approaches are useful approaches to solve the problem for practical purposes.

Let the actual mesh router placement area be a two-dimensional continuous area with width ($W\in R_{+}$) and height ($H\in R_{+}$). In this area, we place a number of mesh routers and mesh clients with fixed positions. Given a network graph $G_{MR}=\left(V_{MR},E_{MR}\right)$ of mesh routers and the network graph $G_{MC}=\left(V_{MC},E_{MC}\right)$ of mesh clients, there are $n=\#V_{MR}$ mesh routers with their own communication range. The positions of mesh routers are not pre-determined and they should be computed. There are $m=\#V_{MC}$ mesh clients located in arbitrary points of the considered area. Each mesh router ($V_{MR}=\left\{V_{MRi}\right\}$) in the graph ($G_{MR}$) is a triple $V_{MR\forall i}=<x,y,r>$ representing the two-dimensional location point $V_{MR\forall i}\left(x\right)\in W$ and $V_{MR\forall i}\left(y\right)\in H$. The mesh routers have a range of radius within which wireless communication is possible ($V_{MR\forall i}\left(r\right)$ is the radius of the communication range). Each mesh client ($V_{MC}=\left\{V_{MCk}\right\}$) in the graph ($G_{MC}$) is a double $V_{MC\forall k}=<x,y>$ representing the two-dimensional location point $V_{MC\forall k}\left(x\right)\in W$ and $V_{MC\forall k}\left(y\right)\in H$. The edge between each mesh router $E_{MR}=\left\{E_{MRi}\right\}$ is calculated by Equation (Equation 1) and the edge between the mesh client and mesh router $E_{MC}=\left\{E_{MCk}\right\}$ is calculated by Equation (Equation 2).
(1)$$\begin{array}{c} E_{MRi}=\left\{\begin{array}{cc} <i,j> & if\;\sqrt{|V_{MRi}\left(x\right)-V_{MRj}\left(x\right){|}^{2}+{\left|V_{MRi}\left(y\right)-V_{MRj}\left(y\right)\right|}^{2}}\\ & \;\;\;\;\;\leq V_{MRi}\left(r\right)+V_{MRj}\left(r\right),\\ 0 & otherwise. \end{array}\right. \end{array}$$
(2)$$\begin{array}{c} E_{MCk}=\left\{\begin{array}{cc} <k,j> & if\;\sqrt{|V_{MCk}\left(x\right)-V_{MRj}\left(x\right){|}^{2}+{\left|V_{MCk}\left(y\right)-V_{MRj}\left(y\right)\right|}^{2}}\\ & \;\;\;\;\;\leq V_{MRj}\left(r\right),\\ 0 & otherwise. \end{array}\right. \end{array}$$

The mesh router placement problem is a bi-objective optimization problem to maximize (1) the network connectivity of WMNs and (2) the coverage of mesh clients. The mesh router placement problem can be seen as a variation of the *p*-median problem. We consider SGC and NCMC as optimization metrics for the mesh router placement problem. Network connectivity of mesh routers is measured by the SGC of the resulting WMN graph, while the coverage of mesh clients is simply the NCMC that fall within the radius (communication range) coverage of at least one mesh router. It should be noted that network connectivity and coverage are among the most important metrics in WMNs and they directly affect the network performance. The adjacency matrices $A_{MR}={\left[a_{MRij}\right]}_{n\times n,(i,j)}$ in Equation (Equation 3) and $A_{MC}={\left[a_{MCkj}\right]}_{m\times n,(k,j)}$ in Equation (Equation 4) of mesh routers and mesh clients are considered to calculate SGC and NCMC.
(3)$$\begin{array}{c} a_{MRij}=\left\{\begin{array}{cc} 1 & if\;\left(\exists <i,j>\in E_{MRi}\right)\cap \left(\exists <j,i>\in E_{MRj}\right),\\ 0 & otherwise. \end{array}\right. \end{array}$$
(4)$$\begin{array}{c} a_{MCkj}=\left\{\begin{array}{cc} 1 & if\;\left(\exists <k,j>\in E_{MCk}\right)\cap \left(\exists <j,k>\in E_{MCj}\right),\\ 0 & otherwise. \end{array}\right. \end{array}$$

The island of connected mesh routers represented by $A_{MR}$ is called the component and there are s≤n components. A set of components $C=\{C_{s}\}$ and $C_{\forall s}\subseteq V_{MR}$ show the connected mesh routers. If $\#C_{\forall s}=\#V_{MR}$ then $C_{\forall s}$ is always called SGC because all (*n*) mesh routers are connected and $C_{\forall s}$ has the largest size of component. If ${⋂}_{k}{⋃}_{j}a_{MCkj}$ = 1, then NCMC is maximal because all (*m*) mesh clients are covered by at least one mesh router. Therefore, by maximizing $C_{\forall s}$ and $\sum_{k}\sum_{j}a_{MCkj}$, we maximize SGC and NCMC.

## 4. Proposed Approach

In this section, we discuss the proposed system. We have considered a bi-objective optimization in which we first maximize the network connectivity of the WMN (the maximization of the SGC) and then the maximization of the NCMC without worsening the value of the SGC.

### 4.1. CCM for Mesh Router Placement Optimization

In this subsection, we describe the operation of the randomly generating mesh routers method for CCM.

In Figure 1 and Algorithm 1 we show the flowchart and pseudo code, respectively, of the randomly generating mesh routers by CCM. In Figure 2 and Algorithm 2 we show the flowchart and pseudo code of the CCM, respectively. The initial placement of mesh routers is done by generating random numbers for mesh router placement, by setting the number of mesh routers, initializing the mesh router linked list, and setting the current number of mesh routers *i* to 0. Next, it generates a mesh router [*i*] at a random placement in the considered area as the 0-th mesh router and the mesh router [*i*] is added to the linked list of mesh routers. After that, *i* is increased by 1 and a mesh router [*i*] is generated at a random placement as the *i*-th mesh router. If the SGC is not increased, then the mesh router [*i*] is removed and a mesh router [*i*] is generated at a random placement again. If the SGC is increased, the mesh router [*i*] is added to the linked list of mesh routers. This process is repeated until *i* is equal to the number of mesh routers n−1. In this way a group of mesh routers can be generated that is connected to other mesh routers as the linked list of mesh routers without Depth First Search (DFS) [35]. However, this method does not take into account the coverage of mesh clients by the mesh routers. Therefore, we use the CCM for deriving the NCMC.
**Algorithm 1** Randomly generating mesh routers method1:Set number of mesh routers *n*.2:Randomly place mesh router [0] in a considered area.3:i←1.4:**while**i<n**do**5:   Randomly place mesh router [0] in a considered area.6:   **if** SGC is maximized **then**7:     i←i+1.8:   **else**9:     Delete mesh router [*i*].10:   **end if**11:**end while**12:Return linkedmeshroutersplacementlist.

**Algorithm 2** CCM
1:Set $I_{max}$.2:i←0.3:S← Algorithm 1.4:$S^{\prime }\leftarrow S$.5:
**while**

$i<I_{max}$

**do**
6:   NCMC(S)←NCMCofS.7:   $NCMC(S^{\prime })\leftarrow$$NCMC\;of\;S^{\prime }$.8:   **if** $NCMC\left(S^{\prime }\right)\geq NCMC\left(S\right)$ **then**9:     $NCMC\left(S\right)\leftarrow NCMC\left(S^{\prime }\right)$.10:     $S\leftarrow S^{\prime }$.11:   **end if**12:   i←i+1.13:   $S^{\prime }\leftarrow$ Algorithm 1.14:   $NCMC(S^{\prime })\leftarrow 0$.15:
**end while**
16:Return *S*.


The process of CCM is as follows. The CCM performs an initialization process by loading the cover list of the mesh client by the mesh router, initializing the optimal linked list of mesh routers *S*, the current linked list of mesh routers $S^{\prime }$, the number of loops in CCM $I_{max}$, and setting the current loop number *i* to 0. Additionally, it initializes the *S* with the linked list of mesh routers decided by the randomly generating mesh router placement method. Next, 1 is added to *t* and the $S^{\prime }$ is generated with the linked list of mesh routers decided by the randomly generating mesh routers method. If the NCMC of $S^{\prime }$ ($NCMC(S^{\prime })$) is greater than the NCMC of *S* (NCMC(S)), then update the optimal linked list of mesh routers with the current linked list of mesh routers. This process is repeated until *i* is equal to the $I_{max}$. Hence, the CCM can decide the placement of mesh routers with the highest NCMC by repeating the randomly generating mesh router placement and deriving NCMC.

### 4.2. CCM-Based SA for Mesh Router Placement Optimization

In this subsection, we describe the CCM-based SA for the mesh router placement problem. In Figure 3 and Algorithm 3 we show the flowchart and pseudo code of the CCM-based SA, respectively. The SA prevents the stacking of the algorithm in local optima. This is a local search algorithm and it is very simple and robust for solving various optimization problems. The SA is inspired by the process of cooling metal materials. It repeatedly searches for neighboring solutions while transitioning states. The best solution during iteration is defined as *S* and the current solution, which is a neighboring solution, is defined as $S^{\prime }$. If the $S^{\prime }$ is worse than the *S*, it derives the optimal solution by transitioning states according to the State Transition Probability (STP). The SA requires the solution evaluation and the temperature value to decide the STP. The evaluations of the mesh router placements (δEval), temperature (*T*), and STP in our proposed approach are shown in Equations (Equation 5)–(Equation 7), respectively.
(5)$$\begin{array}{c} \delta Eval\;\leftarrow NCMC\left(S\right)-NCMC\left(S^{\prime }\right) \end{array}$$
(6)$$\begin{array}{c} T\;\leftarrow \;T_{min}+\left(T_{min}-T_{max}\right)\times \left(\frac{i}{I_{max}}\right) \end{array}$$
(7)$$STP\;\leftarrow \;e^{-\frac{\alpha \times \delta Eval}{T}}$$

The CCM-based SA performs neighborhood search and optimization by randomly changing the placement of one mesh router. The proposed approach updates the optimal solution only when the SGC is maximized and the NCMC of $S^{\prime }$ (NCMC($S^{\prime }$)) is larger than the NCMC of *S* (NCMC(*S*)). Additionally, if SGC is at maximum and NCMC decreases, it will be updated with a probability of STP[%]. *T* represents the cooling schedule at STP and decreases its value from $T_{max}$ to $T_{min}$ depending on the current number of loops *i* at $I_{max}$, which indicates the number of loops in SA. The α is a constant in SGC and the higher the α, the lower the STP.

In the following, we describe the operation of the CCM-based SA. First, the CCM-based SA performs the initialization process by loading the list of mesh clients and setting the InitialandFinaltemperatures, $T_{max}$, and *i* to 0. Additionally, the solution of CCM is set as the current optimal solution *S* and the current solution $S^{\prime }$. Next, the $index_{random}$ is generated at a random value in the range of 0 to the numberofmeshrouters−1. Then, the placement list $S^{\prime }$ [$index_{random}$] is changed at random. If the SGC of the $S^{\prime }$ is maximized, then the SA process is performed. In the SA process, the *r* is generated at random in the range of 0.0 to 100.0. If the $e^{-\frac{\alpha \times \delta Eval}{T}}$ is greater than or equal to 1.0, or if the $e^{-\frac{\alpha \times \delta Eval}{T}}$ is greater than *r*, the optimal linked list of mesh routers *S* is updated to the list of $S^{\prime }$ and 1 is added to *i*. Otherwise, SA does not update the list of *S* and sets $S^{\prime }$ to *S*. This process is repeated until *i* equals the $I_{max}$.
**Algorithm 3** CCM-based SA1:Set $I_{max},\;T_{min},\;T_{max},\;\alpha$.2:i←0.3:S← Algorithm 2 (Meshclientsplacementlist).4:$S^{\prime }\leftarrow$*S*.5:**while**$i<I_{max}$**do**6:   Randomly choose an index of $S^{\prime }$.7:   Randomly change coordinate of $S^{\prime }$ [chosenindex].8:   **if** SGC is maximized **then**9:     r← Randomly generate in (0.0, 100.0).10:     $\delta Eval\leftarrow NCMC\left(S\right)-NCMC\left(S^{\prime }\right)$.11:     $T\leftarrow T_{min}+\left(T_{min}-T_{max}\right)\times \left(\frac{i}{I_{max}}\right)$.12:     **if** $e^{-\frac{\alpha \times \delta Eval}{T}}\geq 1.0$ **then**13:        $NCMC\left(S\right)\leftarrow NCMC\left(S^{\prime }\right)$.14:        $S\leftarrow S^{\prime }$.15:     **else if**
$e^{-\frac{\alpha \times \delta Eval}{T}}\geq r$
**then**16:        $NCMC\left(S\right)\leftarrow NCMC\left(S^{\prime }\right)$.17:        $S\leftarrow S^{\prime }$.18:     **else**19:        Restore coordinate of $S^{\prime }$ [chosenindex].20:     **end if**21:   **else**22:     Restore coordinate of $S^{\prime }$ [chosenindex].23:   **end if**24:   t←t+1.25:**end while**26:Return *S*.

### 4.3. DECCM-Based SA

In the previous works, the mesh routers were placed in the position to cover target mesh clients that were randomly generated by normal distribution or uniform distribution. Thus, when randomly generating or changing the mesh router placement, the entire area was considered as the target of mesh router placement. However, the mesh router placement optimization should consider the bias possibility or the generation of distant mesh client placements in order to be used in more realistic scenarios. The CCM-based SA generates or changes the placement of mesh routers at random. However, this may affect the connectivity of the mesh routers and increase the computation when considering realistic scenarios. Therefore, we propose the DECCM-based SA for more effective mesh router placement optimization.

In the following, we describe the DECCM-based SA. First, the Voronoi decomposition [36] is performed to separate each area where mesh clients are close to each other in the considered area. Each region separated by Voronoi decomposition is called a Voronoi cell and the generated clusters of mesh clients are connected by edges based on the adjacency of each Voronoi cell. These lines are called DE and this process is performed before the CCM process. The DE is used for restricting the placement of mesh routers at random in CCM. The optimizing of the mesh routers placement by DE-based CCM with SA is called DECCM-based SA. This approach can decide the mesh router placements with higher NCMC by restricting the random placement in scenarios with the bias or distant placement of mesh clients compared with the previous approaches.

## 5. Simulation Results

In this section, we present a comparison study of the presented approaches. The parameters for the simulations are shown in Table 1. We set the placement of mesh clients based on GIS [37]. The considered area for the simulation is a disaster area at Kurashiki Train Station in Kurashiki City, Okayama Prefecture, Japan. The considered area was affected by heavy rain in July, 2018. The mesh clients in the simulation are set in the buildings considered as evacuation areas. We used the QGIS [38], which is a GIS application to visualize geographic information. In addition, we used the shapefiles of buildings from OpenStreetMap (OSM) [39] and the open data released by Kurashiki City [40].

Figure 4 shows the visualization results of the considered area, a Voronoi diagram and DE. In Figure 4a,c, the red points indicate evacuation points. Figure 4a shows the image of the original map. In Figure 4b, the red-filled areas show evacuation areas. Figure 4c,d show the visualization results of Voronoi Edge (VE) and DE derived by the Voronoi decomposition. In the proposed approach, the extracted color pixels from images are converted to evacuation areas and the DE are converted to the coordinate list as information that can be used for the simulation.

Figure 5a shows the extracted and converted coordinates of the evacuation area from Figure 4a, while Figure 5b shows the extracted and converted coordinates of the DE from Figure 4d. The coordinates of the red-filled area in Figure 5a are used as mesh clients. The DECCM-based SA uses the coordinates of the DE in Figure 5b as the placement area. The simulations were performed 100 times for each approach.

Figure 6 shows the NCMC vs. the number of iterations in the case of optimization by DECCM-based SA and CCM-based SA (not using DE). The NCMC of the DECCM-based SA was more stable than the CCM-based SA throughout all iterations. Additionally, the DECCM-based SA can cover more mesh clients than the CCM-based SA. Thus, the DECCM-based SA can perform better optimization than the CCM-based SA.

Figure 7 shows the performance evaluation results of 100 times for each approach using box plots (see Figure 7a,b), while Figure 8 shows the visualization results of CCM (the initial generation of CCM-based SA), CCM-based SA, DECCM (the initial generation of DECCM-based SA), and DECCM-based SA. Figure 8a shows the placement of mesh routers by CCM and is concentrated on a specific area. Figure 8b shows the placement of mesh routers optimized by CCM-based SA and the placement area is spread widely compared with the CCM but does not cover all mesh clients. Figure 8c shows the placement of mesh routers by DECCM and covers many mesh clients. Figure 8d shows the placement of mesh routers optimized by DECCM-based SA, which covers all mesh clients. Table 2 shows the best SGC, average SGC, best NCMC, and average NCMC of the simulation results for each approach.

From the simulation results, we can see that the SGC is always maximized for each approach but the DECCM-based SA has better performance than other approaches, especially in scenarios with biased and distant placement of mesh clients.

## 6. Conclusions

In this paper, we proposed and evaluated the DECCM-based SA approach. The proposed approach was simulated in a GIS-based simulation scenario assuming an evacuation area. Then, we compared the performance with other approached using computer simulations. The simulation results show that the DECCM-based SA has a better performance than other approaches. From the simulation results, we conclude as follows:The proposed DECCM-based SA approach can cover all evacuation areas.The DECCM-based SA was able to cover more mesh clients than the CCM-based SA on average.The visualization results show that the DECCM-based SA has better behavior than other approaches.The DECCM-based SA can improve network connectivity and coverage in WMNs by restricting random placement of mesh clients when there is bias or distance in their placement.

In the future, we would like to compare the performance of our proposed system with other state-of-the-art methods and evaluate the proposed approaches via extensive simulations for different scenarios. We would like to implement new hybrid systems considering different intelligent algorithms and make a comparison study of different systems.

## Figures and Tables

**Figure 1 sensors-23-01050-f001:**
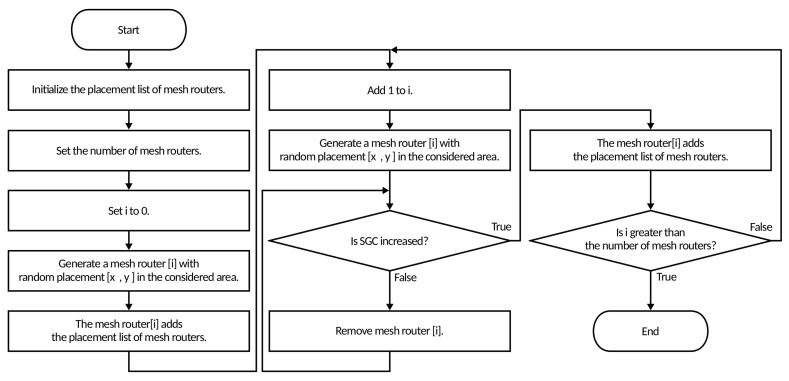
Flowchart for method of randomly generating mesh routers.

**Figure 2 sensors-23-01050-f002:**
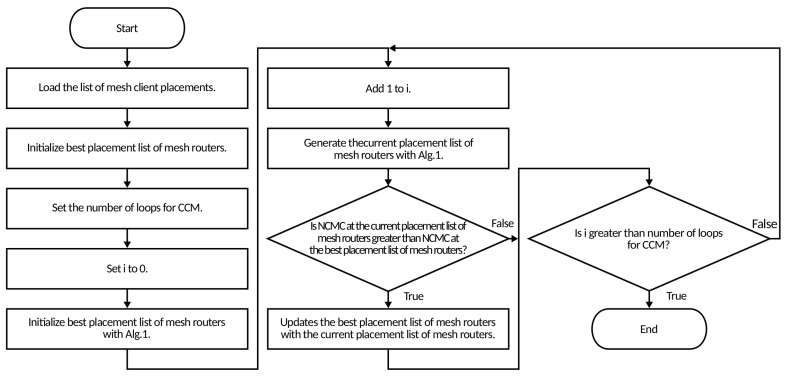
Flowchart of CCM.

**Figure 3 sensors-23-01050-f003:**
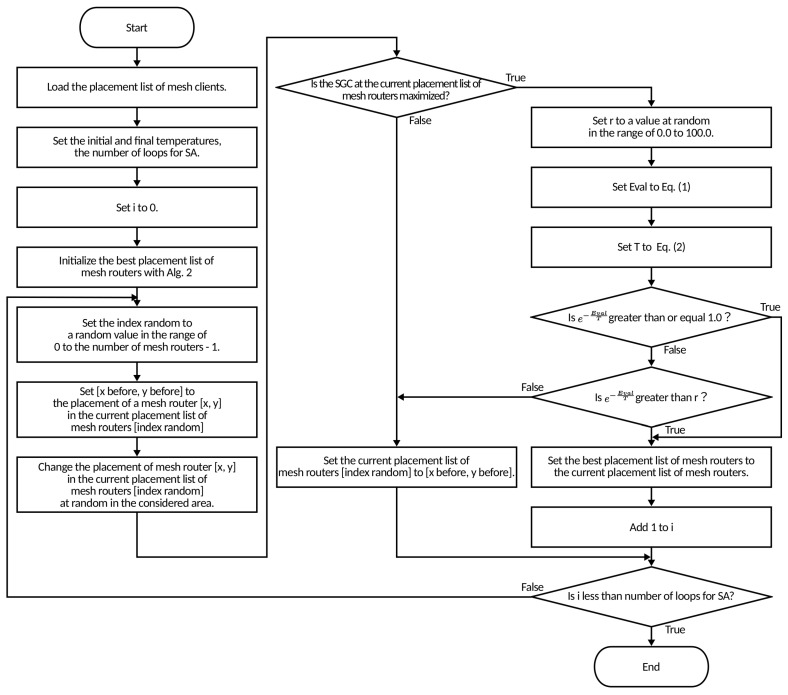
Flowchart of CCM-based SA.

**Figure 4 sensors-23-01050-f004:**
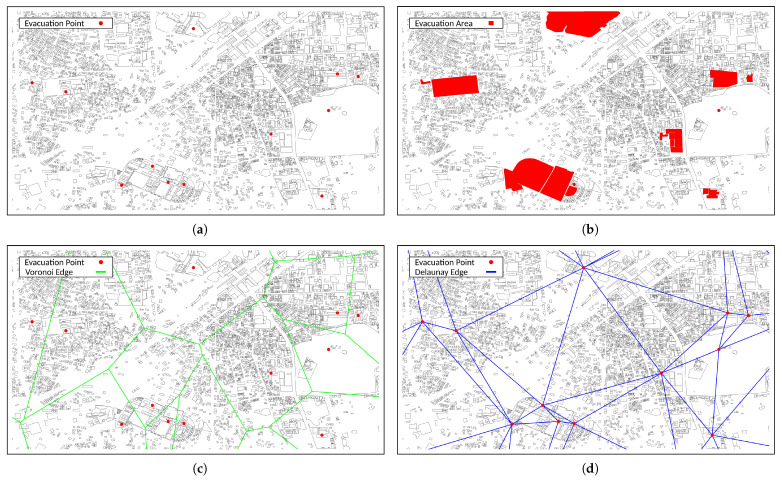
Visualizing the problem area. (**a**) Original map. (**b**) Evacuation area. (**c**) Voronoi diagram. (**d**) Delaunay edge.

**Figure 5 sensors-23-01050-f005:**
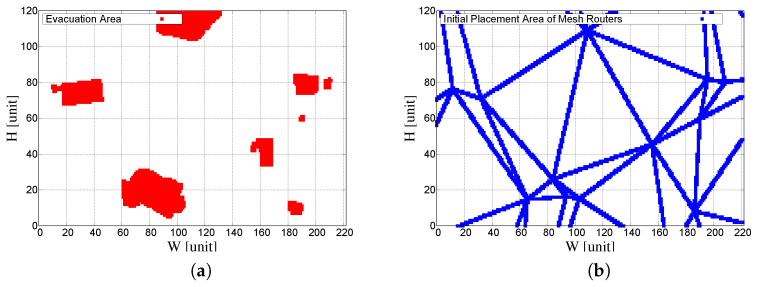
Converted information for the proposed system. (**a**) Converted evacuation area. (**b**) Converted DE.

**Figure 6 sensors-23-01050-f006:**
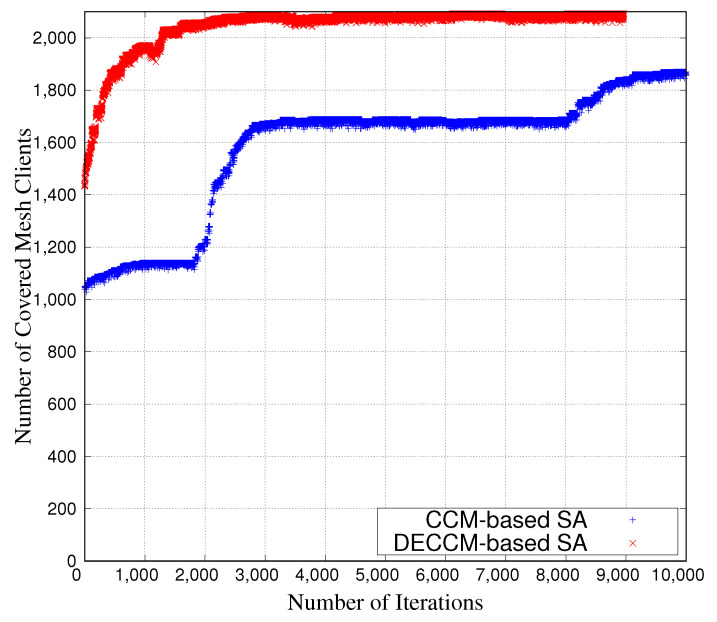
NCMC vs. number of iterations.

**Figure 7 sensors-23-01050-f007:**
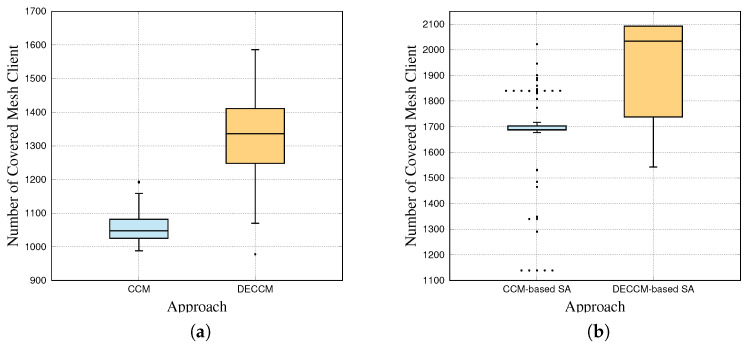
Box-plot results. (**a**) Results of CCM. (**b**) Results of SA.

**Figure 8 sensors-23-01050-f008:**
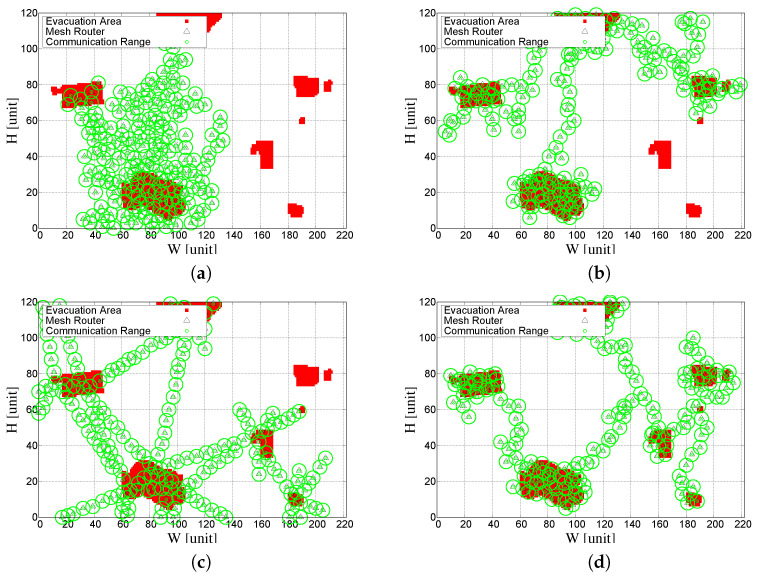
Visualization results. (**a**) Result of CCM. (**b**) Result of CCM-based SA. (**c**) Result of DECCM. (**d**) Result of DECCM-based SA.

**Table 1 sensors-23-01050-t001:** Parameters and values for simulations.

Parameters	Values
*W* (width of considered area)	222 [unit]
*H* (hight of considered area)	120 [unit]
*n* (number of mesh routers)	192
*r* (radius of the mesh routers	4 [unit]
communication range)	
*m* (number of mesh clients)	2092
$I_{max}$ for CCM (number of loops for CCM)	2000 [times]
$I_{max}$ for SA (number of loops for SA)	10,000 [times]
$T_{max}$ (initial temperature)	100 [unit]
$T_{min}$ (final temperature)	1 [unit]

**Table 2 sensors-23-01050-t002:** Simulation results.

Method	Best SGC	Average SGC	Best NCMC	Average NCMC [%]
CCM	192	192	1194	50.540
CCM-based SA	192	192	2022	79.893
DECCM	192	192	1586	63.675
DECCM-based SA	192	192	2092	93.426

## Data Availability

Not applicable.

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
