# Peer review of "A Delaunay Edges and Simulated Annealing-Based Integrated Approach for Mesh Router Placement Optimization in Wireless Mesh Networks"

_sensors, 2023, doi:10.3390/s23031050_

Round 1

Reviewer 1 Report

The paper deals with simulation results that are based on methods that are not suitable for WSN in a realistic approach. for instance, the radius as a metric is not always appropriate for ranges and propagation of electro-magnetic waves.

The presented method could be used in a set-up process that should be adapted in operation taking also into account self healing, processing times ....

An implementation of the method in wireless devices with a proof of concept is mandatory. 

Author Response

Thank you very much for your suggestions and comments.

------------------------------------------------------

Q1:

 The paper deals with simulation results that are based on methods that are not suitable for WSN in a realistic approach. for instance, the radius as a metric is not always appropriate for ranges and propagation of electro-magnetic waves.

Q2:

 An implementation of the method in wireless devices with a proof of concept is mandatory.

Reply to Q1 and Q2:

 Our research is concentrated on mesh router placement optimization in Wireless Mesh Network (WMN) not in Wireless Sensor Networks (WSNs).

 In this study, the communication range of mesh routers is units. The units have been added in Page 9, Section 5: Simulation Results, Table 1 Parameters and values for simulations. The considered area is about 11065 [km^2] and the radius of the communication range of the mesh routers is about 80 [m].

------------------------------------------------------

Q3:

 The presented method could be used in a set-up process that should be adapted in operation taking also into account self healing, processing times ....

Reply to Q3:

 We will consider this suggestion in our future work.

Reviewer 2 Report

The authors proposed a Coverage Construction Method (CCM) that optimizes mesh router placement and presented an integrated optimization approach that combines Simulated Annealing (SA) and Delauney Edges (DE) in CCM to improve the performance of mesh router placement optimization. The detailed comments are as follows.

The problem definition is not formal enough and needs to be redefined in a mathematical format.

The contributions of the paper are not clear. What are the differences between the DECCM-based SA and previous works? The experimental department only compared with the CCM-based SA approach. The authors need to compare with more state-of-art methods in their experiments. 

The technical improvement of the method is not enough.

It is better to present methods in pseudocode rather than flowcharts.

The typesetting of the article could be improved.

The definitions of Eq. 1 and 2 are not mathematically formal, please redefine them.

Author Response

 Thank you very much for your valuable suggestions and comments.

-----------------------------------------------------------------

Q1:

 The problem definition is not formal enough and needs to be redefined in a mathematical format.

Reply to Q1:

 The definition of the mesh router placement problem has been redefined by the mathematical way in Pages 3 and Pages 4, Section 2: Mesh Router Placement Problem, from line 118 to line 149).

-----------------------------------------------------------------

Q2:

 The contributions of the paper are not clear. What are the differences between the DECCM-based SA and previous works? The experimental department only compared with the CCM-based SA approach. The authors need to compare with more state-of-art methods in their experiments.

Q3:

 The technical improvement of the method is not enough.

Reply to Q2 and Q3:

 The contributions of this work are shown in Page 2, from line 51 to line 60. In this research, we have compared 3 approaches. But we will consider to compare the proposed approach with state-of-the-art methods in our future work. We added in the last section, Page 12, Section 6 Conclusions, from line 204 to line 205, the following text:

 In the future, we will compare the performance of our proposed system with other state-of-art methods.

-----------------------------------------------------------------

Q4:

 It is better to present methods in pseudocode rather than flowcharts.

Q5:

 The typesetting of the article could be improved.

Reply to Q4 and Q5:

 We have added pseudo codes of Algorithm 1, Algorithm 2 and Algorithm 3 from Page 5 to Page 7, Section 4: Proposed Approach.  

-----------------------------------------------------------------

Q6:

 The definitions of Eq. 1 and 2 are not mathematically formal, please redefine them.

Reply to Q6:

 Eq. (1) and Eq. (2) have been redefined in mathematical form. In the current version of the paper, they are Eq. (5) and Eq. (6). See Section 4 Proposed Approach, Page 7.

Reviewer 3 Report

In this paper, author has presented the coverage construction method for optimization of mesh router placement. The results show the better performance of the optimization of mesh router placement. The method presented in the paper is useful for better optimization of mesh router placement in wireless mesh networks. Following are the suggestions to the authors:

1.       The title of the paper should be shortened and use of abbreviations in the title should be avoided as ‘DECCM’ and ‘SA’ are not very common abbreviations.

2.       In the abstract, please include about the performance of the proposed algorithm, why the placement of mesh routers is optimum using the proposed method? Please highlight about this in the abstract.

3.       In keywords, please remove bracket ‘(‘ before ‘Delauney’.

4.       The language for structure of the paper at the end of section 1 i.e. ‘Introduction’ should be improved. The sentences used for section 2 to section 4 should be rephrased.

5.       The quality of Figure 1 and Figure 2 should be improved.

6.       Please specify the units of the parameters in Table 1.

7.       There are some typos and grammatical mistakes, please proofread the paper.

Author Response

 Thank you very much for your suggestions and comments.

-----------------------------------------------------------------

Q1:

 The title of the paper should be shortened and use of abbreviations in the title should be avoided as ‘DECCM’ and ‘SA’ are not very common abbreviations.

Reply to Q1:

 The title of the paper has been revised as follows:

 An Integrated Approach for Mesh Router Placement Optimization in Wireless Mesh Networks Considering Delaunay Edges and Simulated Annealing

-----------------------------------------------------------------

Q2:

 In the abstract, please include about the performance of the proposed algorithm, why the placement of mesh routers is optimum using the proposed method? Please highlight about this in the abstract.

Reply to Q2:

 The following text has also been added to the Abstract and Conclusions:

 Page 1, Abstract, from line 13 to line 14:

 Also, the DECCM-based SA approach covers more mesh clients than the CCM-based SA approach on average and can improve network connectivity of WMNs.

 Page 12, Conclusion, from line 297 to line 299:

 The DECCM-based SA can improve network connectivity and coverage in WMNs by restricting random placement of mesh clients when there is bias or distance in their placement.

-----------------------------------------------------------------

Q3:

 In keywords, please remove bracket ‘(‘ before ‘Delauney’.

Reply to Q3:

 We have removed the bracket as you suggested.

-----------------------------------------------------------------

Q4:

 The language for structure of the paper at the end of section 1 i.e. ‘Introduction’ should be improved. The sentences used for section 2 to section 4 should be rephrased.

Reply to Q4:

 We revised the last section of the Introduction as you suggested, Page 2, from line 61 to line 63.

-----------------------------------------------------------------

Q5:

 The quality of Figure 1 and Figure 2 should be improved.

Reply to Q5:

 The figures have good quality.

-----------------------------------------------------------------

Q6:

 Please specify the units of the parameters in Table 1.

Reply to Q6:

 We revised Table 1. For simulations, we use units as you can see in Fig. 5 and Fig. 8.

-----------------------------------------------------------------

Q7:

 There are some typos and grammatical mistakes, please proofread the paper.

Reply to Q7:

 The paper was revised by a native speaker.

-----------------------------------------------------------------

Round 2

Reviewer 2 Report

The newly submitted manuscript has made certain improvements. I still have some other comments.

The language and writing quality of the manuscript can be improved. Please proofread.

The format of references should be unified. There are many errors and style inconsistencies.

Author Response

Thank you very much for your valuable suggestions and comments.

-----------------------------------------------------------------

Q1: The language and writing quality of the manuscript can be improved. Please proofread.

Reply to Q1:

I checked the whole paper carefully and corrected some misprints.

-----------------------------------------------------------------

Q2: The format of references should be unified. There are many errors and style inconsistencies.

Reply to Q2:

I checked and revised references to be in the same format as reviewer suggested.

-----------------------------------------------------------------

Reviewer 3 Report

Paper still needs improvements. Following are the suggestions to the authors:

1.       The title may be improved either as ‘A Delaunay Edges and Simulated Annealing-Based Integrated Approach for Mesh Router Placement Optimization in Wireless Mesh Networks’ or ‘An Integrated Approach for Mesh Router Placement Optimization in Wireless Mesh Networks’.

2.       There are still some mistakes such as ‘In Section 2 is presented the related work.’ is wrong, it should be written as either ‘The related work is presented in Section 2.’ or ‘Section 2 presents the related work.’. The author should proofread the paper carefully.

3.       Page 3: ‘Line-of-Sight (NLoS) environment’ should be corrected as ‘Line-of-Sight (LoS) environment’.

4.       Page 4: There is one extra bracket in (Eq. 3)) and (Eq. 4)), please correct.

5.       Please include the information of X-axis and y-axis in Fig. 5 and Fig. 8.

6.       In conclusion section, future work should be improved. Author should discuss about the further improvements in the proposed work.

7.       There are many grammatical mistakes in the paper.

Author Response

Thank you very much for your suggestions and comments.

-----------------------------------------------------------------

Q1: The title may be improved either as ‘A Delaunay Edges and Simulated Annealing-Based Integrated Approach for Mesh Router Placement Optimization in Wireless Mesh Networks’ or ‘An Integrated Approach for Mesh Router Placement Optimization in Wireless Mesh Networks’.

Reply to Q1:

I revised the title as suggested by reviewer.

-----------------------------------------------------------------

Q2: There are still some mistakes such as ‘In Section 2 is presented the related work.’ is wrong, it should be written as either ‘The related work is presented in Section 2.’ or ‘Section 2 presents the related work.’. The author should proofread the paper carefully.

Reply to Q2:

I revised this part as suggested by reviewer (see page 2, line 61). Also, I made a careful check of the whole paper.

-----------------------------------------------------------------

Q3: Page 3: ‘Line-of-Sight (NLoS) environment’ should be corrected as ‘Line-of-Sight (LoS) environment’.

Reply to Q3:

I revised this as reviewer suggested (see page 3, line 102).

-----------------------------------------------------------------

Q4: Page 4: There is one extra bracket in (Eq. 3)) and (Eq. 4)), please correct.

Reply to Q4:

 I corrected this by revising the sentence (see page 3, line 141 to line 142).

-----------------------------------------------------------------

Q5: Please include the information of X-axis and y-axis in Fig. 5 and Fig. 8.

Reply to Q5:

I included the information of X-axis and y-axis in Fig. 5 and Fig. 8.

-----------------------------------------------------------------

Q6: In conclusion section, future work should be improved. Author should discuss about the further improvements in the proposed work.

Reply to Q6:

I revised the future work (see page 12, line 291 to line 294).

-----------------------------------------------------------------

Q7. There are many grammatical mistakes in the paper.

Reply to Q7:

I checked the whole paper carefully and corrected some misprints.

-----------------------------------------------------------------